# Changes in Fish Assemblage Structure after Pen Culture Removal in Gehu Lake, China

Xiaoliang Ren [1,†], Shulun Jiang [2,†], Long Ren [2], Yidong Liang [1], Di'an Fang [2,3] and Dongpo Xu [2,3,*]

1 College of Fisheries and Life Science, Shanghai Ocean University, Shanghai 201306, China
2 Key Laboratory of Freshwater Fisheries and Germplasm Resources Utilization, Ministry of Agriculture and Rural Affairs, Freshwater Fisheries Research Center, Chinese Academy of Fishery Sciences, Wuxi 214081, China
3 Wuxi Fisheries College, Nanjing Agricultural University, Wuxi 214081, China
* Correspondence: xudp@ffrc.cn; Tel.: +86-0510-8555-9845
† These authors contributed equally to this work.

**Abstract:** The removal of the net enclosure has been used as a lake management strategy in various regions of China as ecological development is given more attention. Nevertheless, little is known about the substantive impact of this measure on fish communities in inland lakes. To this end, the fish community composition and structural features after the removal of the net enclosure in Gehu Lake were explored and evaluated in this study from 2021 to 2022 and compared to the investigation before the net enclosure removal from 2017 to 2018. Belonging to 7 orders, 10 families, and 46 species, a total of 17,151 fish were collected, with pelagic, sedentary, and omnivorous species dominating. In comparison, the number of species increased by 10 after removal, and fish alpha diversity increased. The Index of Relative Importance (IRI) revealed that the composition of dominating species remained constant, including *Coilia nasus*, *Hypophthalmichthys molitrix*, and *Hypophthalmichthys nobilis*; *Parabramis pekinensis*, *Megalobrama amblycephal*, and *Culter mongolicus* were upgraded from common species to general species. *Culter alburnus*, *Hemiculter bleekeri*, and *Pseudobrama simoni* were downgraded from general species to common species. *Elopichthys bambusa* had become a common species (IRI = 109.35), which was not discovered before removal. According to the hierarchical clustering (HC) and non-metric multidimensional sequencing (NMDS), the fish community of the northern reserve was highly aggregated. As the Abundance Biomass Comparison (ABC) curve and biodiversity index indicated, the fish community structure of the whole lake was in a state of moderate anthropogenic disturbance with reduced stability, while that of the northern reserve was in a state of light anthropogenic disturbance with greater stability. The number of fish species increased in this survey compared to the period before removal, species and dominant species composition altered dramatically, and total lake stability declined. This study demonstrates that the fish diversity in Gehu Lake increased after the removal of the net enclosure. Meanwhile, the stability of the fish community structure was decreased temporarily. Lake restoration is a long-term process, and the underlying impact of the removal of the net enclosure still requires continuous monitoring and further studies.

**Keywords:** Gehu Lake; fish assemblage structure; composition; removal of net enclosures; differences

## 1. Introduction

Pen culture refers to intensive aquaculture practice by engineering measures such as enclosing, blocking, and isolating a specific region of water in lakes and reservoirs [1]. After the 1950s, environmental damage and climatic changes decreased natural fishery resources in lakes worldwide, which drove the rapid development of lake aquaculture fisheries with limited fishery resources. Efficient intensive aquaculture techniques, such as pen culture, were gradually developed in some countries [2]. In China, in order to suit the demands of social development and aquatic protein, pen culture incrementally thrives to benefit fishing

yield in the middle and lower Yangtze River [3,4], and also plays an active role in ecology as a lake management technique [5,6].

Gehu Lake, one of the five major lakes in Jiangsu Province, China, with a total size of 166 km$^2$ and a storage capacity of $1.74 \times 10^8$ m$^3$, not only plays an essential role in flood control, water supply for inhabitants, ecological control, and so on, but it also serves as an important aquaculture base for people, bringing tremendous economic benefits. In the early 1980s, Gehu Lake implemented pen culture earlier in China to boost fishing productivity. By the turn of the century, pen culture had expanded to an area of more than 60 km$^2$. Increasingly polluted by nitrogen and phosphorus nutrients, the ecological conditions of Gehu Lake worsened year by year, with the rapid urbanization and the speedy growth of industry and agriculture [7], which led to a dramatic decrease in the aquatic organisms within the lake, high content of nutrient salt in the surface sediment [8], and lowered the ecosystem's stability [9]. As a result of this fact, Gehu Lake has degraded from grass-type to algal-type [10]. The total area of pen culture in Gehu Lake was approximately 15.3 km$^2$ at the time of the fish resource survey in 2017–2018. (Figure 1). Chinese lake fisheries need to transition to a green ecological approach in order to achieve sustainable development. In April 2019, pen culture facilities in Gehu Lake were entirely dismantled under the guideline of Chinese Green Ecological Fisheries [11]. The local government established an ecological restoration area in the center of Gehu Lake in the same year.

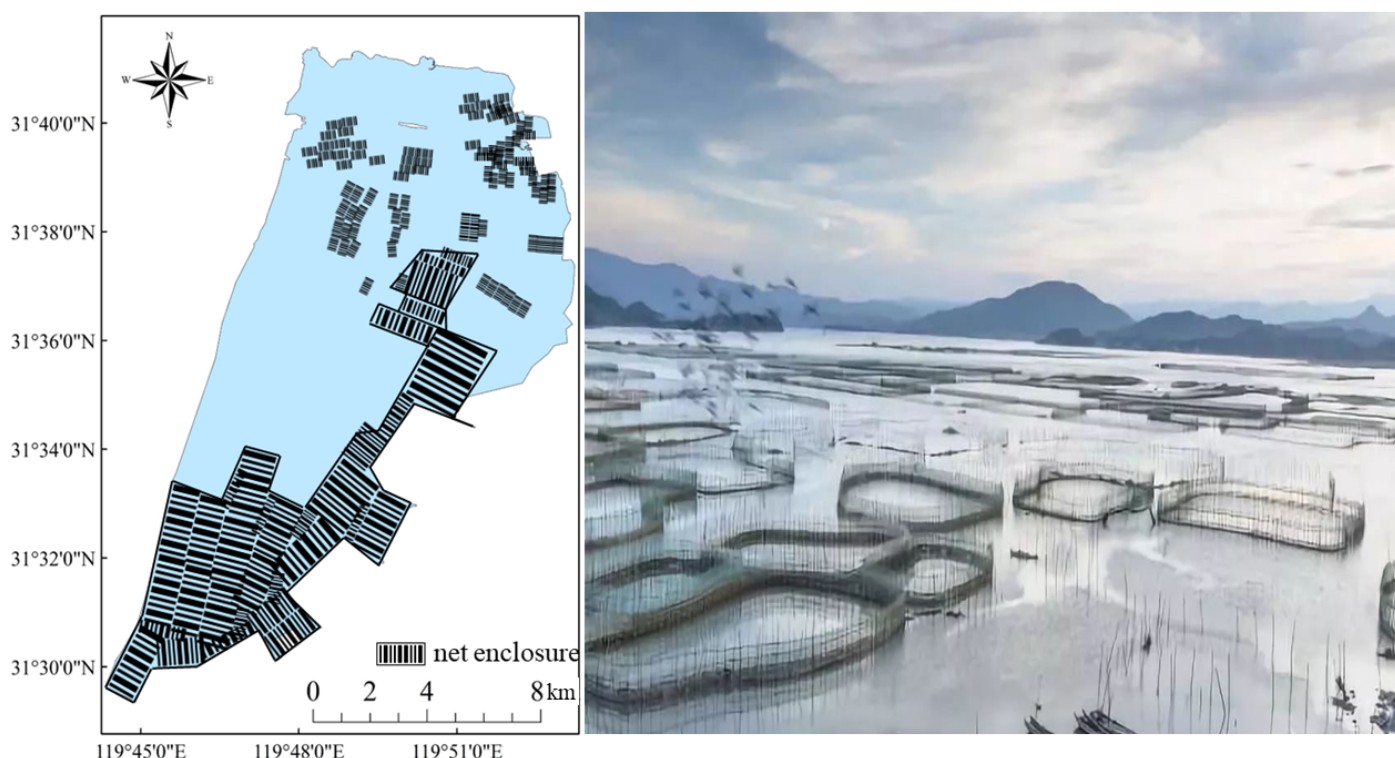

**Figure 1.** Schematic diagram of the pen culture region in Gehu Lake in 2017–2018. The photo of the pen culture facilities is from https://m.sohu.com/a/271925579_649720?_trans_=010004_pcwzy (accessed on 29 October 2018).

Removing the net enclosure will affect the ecological stability of the waters and the structure of the fish community in lakes. According to limited studies, after the removal of the net enclosure, the barrier effect disappeared in the lake, and the increase in migratory and semi-migratory fish, as well as the residual bait in the sediment of the farming area, will increase the $\alpha$-diversity of fish [12]. The water quality, sediment, and fish community structure of the east Taihu Lake were in a relatively stable state before the removal. Nevertheless, the fish community structure turned unstable after the removal, and the community

structure was changed and dominated by fast-growing and small-individual fish [13]. Fish are the top consumers in aquatic ecosystems, and their communities have different ecological, morphological, and behavioral adaptations to their natural habitats. Additionally, fish communities are effective indicators for evaluating aquatic ecosystems [14,15], are important components of aquatic ecosystems, and they play an important role in the structural functions and system stability [16–18]. The composition and structure of the fish community in the lake will be changed to some extent after the net enclosure is removed. Until now, little research on the changes in fish community structure before and after the removal of the net enclosure in lakes has been documented, with only the east Taihu Lake and Huayang Lake being reported in China [12,13]. Therefore, the analysis of the fish community changes before and after the removal of the net enclosure in the Gehu Lake aquatic ecosystem could provide significant ecological indication and essential data to support Gehu Lake's development, conservation, and management. Furthermore, it also has reference values for evaluating the ecological effects of the net enclosure removal in other lakes.

In this study, the fish community composition of Gehu Lake after the removal of the net enclosure was investigated from 2021–2022. The fish species diversity was analyzed by dominant species composition, the Shannon–Wiener diversity index, the Pielou evenness index, and Margalef's richness index, and compared with the result before the removal of the net enclosure (in the period from 2017–2018). The Abundance Biomass Comparison (ABC) curve was used to analyze the changes in fish community status after the removal of the net enclosure, aiming to reveal the changes in the fish community structure and its ecological indication, and provide the scientific basis for lake management and watershed ecological protection.

## 2. Materials and Methods

### 2.1. Study Area

Gehu Lake (119°44′15″~119°52′56″ E, 31°28′19″~31°43′04″ N) is a typical shallow lake, with an average depth of about 1.3 m. It is the second biggest freshwater lake after Taihu Lake in southern Jiangsu, and one of five main lakes in Jiangsu. The Gehu Lake is shaped similar to a long eggplant, with a shallow saucer-shaped body and a flat bottom. It is located east of Taihu Lake, west of Changdang Lake, south of Yixing Jiu Lake, north of the Yangtze River by the Biandan River and Desheng River, and the river and harbor along the lake are crisscrossed with water networks.

### 2.2. Method

This study conducted four surveys at sixteen sampling stations in January, August, November 2017, and May 2018 (Figure 2A) and four surveys at fifteen sampling stations in July, September, November 2021, and January 2022 (Figure 2B), each for seven days. Three multi-mesh monofilament gillnets (length: 125 m, height: 1.5 m, and mesh size: 1.2, 2, 4, 6, 8, 10, and 14 cm) and three fixed series cage pots (length: 10 m, width: 0.4 m, height: 0.4 m, and mesh size: 1.6 cm) were positioned in each site for 24 h. Fish were classified and identified [19,20] immediately after collecting, and the number and weight was measured. Healthy and undamaged fish were returned to their original site. The methodology of the 2017–2018 study is consistent with that described above. The methodology of data analysis is also the same for both time periods.

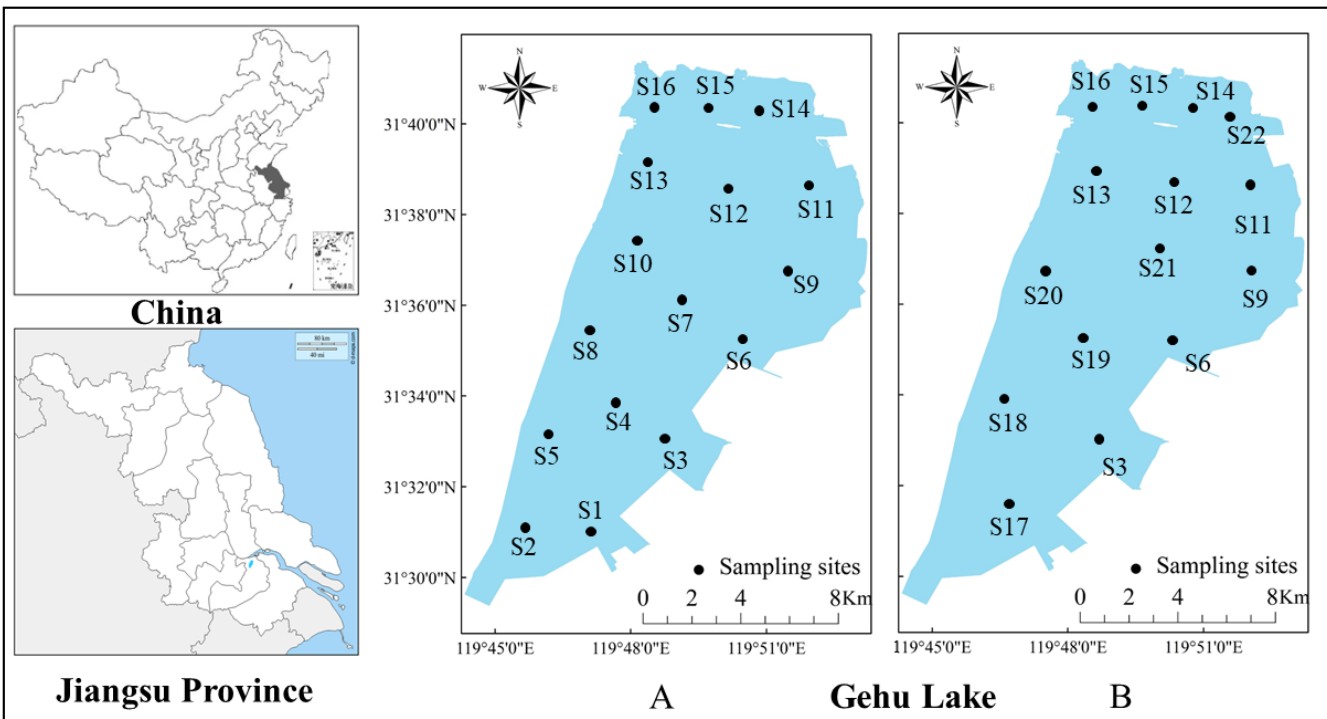

**Figure 2.** The geography of Gehu Lake in Jiangsu, China. Black circles indicate the sampling sites. The sampling sites during 2017–2018 (**A**) are from S1 to S16. The sampling sites during 2021–2022 (**B**) are S3, S6, S9, and S11–S22.

*2.3. Data Analysis*

2.3.1. Ecological Types

According to the habitats and migration, the ecological types of the sampled fish were divided into sedentary (SE), river-lake migratory (RL), estuarine (ES), and river-sea migratory (RS). There are seven types of phytoplanktivores, zoophytoplanktivores, phytobenthivores, zoobenthivores, piscivores, detritivores, and herbivores according to their feeding habits. Classification of fish ecotypes were performed according to references [19,20].

2.3.2. Community Diversity

Margalef's index (*R*), the Shannon–Weiner diversity index (*H'*), and the Pielou evenness index (*J'*) were used to analyze fish community diversity [21–23]. The formulas were as follows:

$$R = (S-1) / \ln N \qquad (1)$$

$$H'_w = -\sum P_i \ln P_i \qquad (2)$$

$$J'_w = H'_w / \ln S \qquad (3)$$

where *S*, *N*, and $P_i$ represent the number of species, the number of individuals, and the proportion of the number of species *i* to the total number of species, respectively.

2.3.3. Relative Importance Index

The Index of Relative Importance (IRI) was used to determine the dominant species in the fish community [24]. The formulas were as follows:

$$\text{IRI} = (N\% + W\%) \times F\% \qquad (4)$$

where *N*% represents the proportion of the number of species *i* to the total number of individuals, *W*% represents the proportion of the weight of species *i* to the total weight of

individuals, and *F*% represents the proportion of the occurrences of species *i* to the total number of surveys.

In this study, IRI classifications were categorized as follows: IRI > 1000 represents a dominant species, $100 \leq$ IRI < 1000 represents a common species, and $10 \leq$ IRI < 100 represents a general species. Dominant and common species are collectively referred to as major species.

### 2.3.4. Community Stability

Abundance Biomass Comparison curves (ABC curves) were used to measure the stability of community structure and the intensity of interference by environmental factors [25]. In a stable or undisturbed state, the biomass dominance curve always lies above the abundance dominance curve. Meanwhile, as the degree of interference increases, and the population becomes moderately disturbed or unstable, the two curves intersect. When the biomass dominance curve remains below the abundance dominance curve, the community is considered severely disturbed or unstable [26].

*W* (the *W* statistic) is the quantitative standard of the ABC curve, and *W* is positive when the biomass dominance curve is above the quantitative dominance curve and negative when the opposite is true. Curves were calculated as follows:

$$w = \Sigma_{i=1}^{s} \frac{(B_i - A_i)}{50(s-1)} \tag{5}$$

where $B_i$ and $A_i$ represent the cumulative percentages of biomass and abundance corresponding to the number of species in the ABC curve, and *S* is the number of species present.

### 2.3.5. Community Clustering Characteristics

Primer v5.0 was used to analyze fish data for the years 2017–2018 and 2021–2022. After removing fish with an individual percentage of less than 1% and performing square root transformation, the Bray–Curtis similarity coefficient matrix was determined according to the "site × number" list. Hierarchical clustering (CLUSTER) and non-metric multidimensional sequencing (NMDS) were then used to analyze the structural characteristics of each fish community. Cluster analysis can be subjectively arbitrary when used to determine similarity levels. It is, therefore, combined with NMDS multi-dimensional scaling to verify the correctness of the results. The strength coefficient (stress test) was then used to test the results of the NMDS analysis. A stress level of <0.2 represents certainty, <0.1 is considered good, and <0.05 is considered a very good representation [27].

## 3. Results

### 3.1. Fish Species Composition and Ecological Types

A total of 17,151 fish were collected from the 15 sampling sites in Gehu Lake, and classified into 46 species, 10 families, and 7 orders by identification. Cypriniformes (thirty-four species, 73.91% of the total species) accounted for the largest number of species, followed by Perciformes (five species, 10.87%), and Siluriformes (three species, 6.52%). Meanwhile, Salmoniformes, Clupeiformes, Beloniformes, and Anguilliformes contained one species each, forming 2.17% of the total species collectively. At the family level, the most numerically abundant species was Cyprinidae (thirty-two species, 57.65% of the total species), followed by Bagridae (three species, 6.25%). Serranidae, Cobitidae, and Gobiidae all have two species (4.17%), and the remaining five families each have only one species. *Odontamblyopus rubicundus* and *Pelteobagrus eupogon* were not discovered in this survey, but were found in 2017–2018 (Table 1).

**Table 1.** Fish species composition between the two periods in Gehu Lake.

| Family | Species | 2017–2018 | 2021–2022 | FFG | EG | NAG |
|---|---|:---:|:---:|:---:|:---:|:---:|
| Engraulidae | *Coilia nasus* | √ | √ | plankt | ES | Nat |
| Cyprinidae | *Abbottina rivularis* | | √ | plankt | SE | Nat |
| | *Hemiculter bleekeri* | √ | √ | herbivores | SE | Nat |
| | *Parabramis pekinensis* | √ | √ | herbivores | RL | Nat |
| | *Hemiculter leucisculus* | √ | √ | detritivores | SE | Nat |
| | *Ctenopharyngodon idella* | √ | √ | herbivores | RL | Nat |
| | *Culter dabryi* | √ | √ | piscivores | SE | Nat |
| | *Acheilognathus macropterus* | √ | √ | plankt | SE | Nat |
| | *Megalobrama skolkovii* | | √ | zoobenthivores | SE | Nat |
| | *Elopichthys bambusa* | | √ | piscivores | RL | Nat |
| | *Pseudolaubuca engraulis* | √ | √ | detritivores | RL | Nat |
| | *Sarcocheilichthys nigripinnis* | √ | √ | zoobenthivores | SE | Nat |
| | *Cultrichthys erythropterus* | √ | √ | plankt | SE | Nat |
| | *Hemibarbus maculatus* | | √ | zoobenthivores | SE | Nat |
| | *Xenocypris davidi* | √ | √ | detritivores | RL | Nat |
| | *Aristichthys nobilis* | √ | √ | detritivores | SE | Nat |
| | *Cyprinus carpio* | √ | √ | plankt | SE | Nat |
| | *Cyprinus carpio × Carassius auratus* | | √ | plankt | SE | Non-Nat |
| | *Hypophthalmichthys molitrix* | √ | √ | phytoplankt | RL | Nat |
| | *Cirrhinus mrigala* | | √ | phytoplankt | RL | Ali |
| | *Pseudorasbora parva* | √ | √ | plankt | SE | Nat |
| | *Culter mongolicus* | √ | √ | plankt | RL | Nat |
| | *Culter alburnus* | √ | √ | plankt | RL | Nat |
| | *Saurogobio dabryi* | | √ | plankt | SE | Nat |
| | *Pseudobrama simoni* | √ | √ | phytobenthivores | RL | Nat |
| | *Paracanthobrama guichenoti* | | √ | zoobenthivores | SE | Nat |
| | *Toxabramis swinhonis* | √ | √ | plankt | SE | Nat |
| | *Megalobrama amblycephala* | √ | √ | herbivores | SE | Nat |
| | *Xenocypris microlepis* | | √ | detritivores | RL | Nat |
| | *Acheilognathus chankaensis* | √ | √ | phytobenthivores | SE | Nat |
| | *Xenocypris macrolepis* | | √ | detritivores | RL | Nat |
| | *Aristichthys nobilis* | √ | √ | plankt | RL | Nat |
| | *Saurogobio dumerili* | √ | √ | plankt | SE | Nat |
| | *Rhodeus fangi* | √ | | phytobenthivores | SE | Nat |
| | *Sarcocheilichthys sinensis* | √ | | zoobenthivores | SE | Nat |
| Bagridae | *Pelteobaggrus nitidus* | √ | √ | plankt | SE | Nat |
| | *Pelteobagrus vachelli* | √ | √ | plankt | SE | Nat |
| | *Pelteobaggrus fulvidraco* | √ | √ | plankt | SE | Nat |
| | *Pelteobagrus eupogon* | √ | | plankt | SE | Nat |
| Anguillidae | *Anguilla japonica* | | √ | zoobenthivores | RS | Non-Nat |
| Serranidae | *Siniperca kneri garman* | | √ | piscivores | SE | Nat |
| | *Siniperca chuatsi* | √ | √ | piscivores | SE | Nat |
| Cobitidae | *Paramisgurnus dabryanus* | | √ | plankt | SE | Nat |
| | *Misgurnus anguillicaudatus* | √ | √ | plankt | SE | Nat |
| | *Cobitis sinensis* | √ | | plankt | SE | Nat |
| Gobiidae | *Taenioides cirratus* | √ | √ | zoobenthivores | SE | Nat |
| | *Rhinogobius giurinus* | √ | √ | zoobenthivores | SE | Nat |
| | *Odontamblyopus rubicundus* | √ | | piscivores | SE | Nat |
| Salangidae | *Protosalanx chinensis* | √ | √ | piscivores | ES | Nat |
| Channidae | *Channa argus* | √ | √ | piscivores | SE | Nat |
| Hemirhamphiade | *Hyporhamphus intermedius* | √ | √ | plankt | ES | Nat |

Notes: FFG. feeding functional group (plankt. zoophytoplanktivores; phytoplankt. phytoplanktivores; herbivores; detritivores; piscivores; zoobenthivores; phytobenthivores). EG. ecological group (SE. settled; ES. estuarine; RL. river-lake migratory; RS. river-sea migratory fish). NAG. native or alien group (Nat. native; non-Nat. Non-native; Ali. alien). "√" indicates presence.

In terms of feeding habits, Gehu Lake was dominated by omnivorous fish (twenty-eight species) in this study, which accounted for 60.87% of the total number of species; eight species more than the data before removal in 2017–2018. Twelve species of carnivorous fish make up 26.09% of the total, and three of them, namely *Elopichthys bambusa*, *Siniperca kneri* Garman, and *Anguilla japonica,* were newly discovered after removal. Phytophagy fish have a minimal number of species, with just six species accounting for 13.04%.

In the case of lifestyle habits, twenty-nine species of SE fish were collected (76.09%) with three additional species after removal. Both surveys included three ES fish species (6.52%), e.g., *Coilia nasus* (lake anchovy), *Protosalanx chinensis*, and *Hyporhamphus intermedius*. Meanwhile, one new species of RS fish was found in this survey (*A. japonica*). Thirteen species of RL fish (28.26%) were collected with four additional species (e.g., *Megalobrama skolkovii*, *E. bambusa*, *Cirrhinus mrigala*, and *Xenocypris microlepis*) for comparison.

### 3.2. Dominant Species

Based on IRI calculated by weight, the number of individuals, and the frequency of catches in Gehu Lake, the dominant species were *C. nasus* (IRI = 9074.44), *Hypophthalmichthys molitrix* (IRI = 3296.75), *Aristichthys nobilis* (IRI = 2766.67), and *Carassius auratus* (IRI = 1010.61). These four dominant species included 14,623 individuals, accounting for 76.95% of the total number, and 85.26% of the total biomass. Moreover, there were 11 common species (Table 2).

**Table 2.** Dominant species composition of fish in Gehu Lake before (2017–2018) and after (2021–2022) removal of the net enclosure.

| Fish (Species) | 2017–2018 IRI | 2021–2022 IRI |
|---|---|---|
| *Coilia nasus* | 7175.80 | 9074.44 |
| *Hypophthalmichthys molitrix* | 5595.01 | 3296.75 |
| *Hypophthalmichthys nobilis* | 1778.82 | 2766.67 |
| *Carassius auratus* | 1112.13 | 1010.61 |
| *Chanodichthys dabryi* | 595.49 | 588.53 |
| *Cyprinus carpio* | 431.63 | 518.28 |
| *Chanodichthys erythropterus* | 495.82 | 473.40 |
| *Parabramis pekinensis* | 26.99 | 298.25 |
| *Megalobrama amblycephala* | 28.61 | 170.22 |
| *Pelteobaggrus nitidus* | 293.32 | 156.89 |
| *Pelteobaggrus fulvidraco* | 371.49 | 126.15 |
| *Toxabramis swinhonis* | 247.95 | 122.29 |
| *Culter mongolicus* | 11.12 | 118.43 |
| *Elopichthys bambusa* | 0 | 109.35 |
| *Hemiculter leucisculus* | 102.45 | 102.60 |
| *Culter alburnus* | 341.43 | 92.40 |
| *Hemiculter bleekeri* | 186.22 | 39.23 |
| *Pseudobrama simoni* | 141.90 | 44.97 |

Notes: IRI. the Index of Relative Importance.

Compared to the survey before removal, the composition of dominant species remained unchanged, The IRI of *C. nasus* and *A. nobilis* rose, but those of *H. molitrix* and *C. auratus* declined. Large changes in common species composition occurred as the IRI of *Parabramis pekinensis*, *Megalobrama amblycephala*, and *Culter mongolicus* increased from 26.99 to 298.25, 28.61 to 170.22, and 11.12 to 118.43, respectively, while those of *Culter alburnus*, *Hemiculter bleekeri* and *Pseudobrama simoni* decreased from 341.43 to 92.40, 186.22 to 39.23, and 141.90 to 44.97, respectively. Most noteworthy, a total of 37 *E. bambusa* were discovered in 11 sites of this survey; nonetheless, they were not found in the 2017–2018 survey. Meanwhile, the IRI had reached 109.35, indicating the level of common species. In conclusion, after the net enclosures were entirely removed, the dominant species composition of the fish community in the whole lake changed considerably (Table 2).

### 3.3. Community Structure and Stability

CLUSTER and NMDS were conducted to analyze the fish populations of capture from 15 sites sampled in Gehu Lake between 2021 and 2022 (Figure 3B). Similarity cluster analysis divided the fifteen sampling sites into three groups at a similarity level of 60.51%. The first group included S14, S15, S16, and S22, the second group included S20, and the third group included the rest of the sites. ANOSIM analysis further revealed that the composition of community structure was significantly different among the clustering groups (R = 0.916, $p < 0.05$). Meanwhile, the stress level was 0.1, suggesting a very good level of representation. The results indicated that S14, S15, S16, and S22 were distributed in the northern part of Gehu Lake, forming distinct aggregation areas. On this basis, the data before removal were further clustered and classified, aiming to compare the clustering before and after the removal of the net closure in the northern part of Gehu Lake.

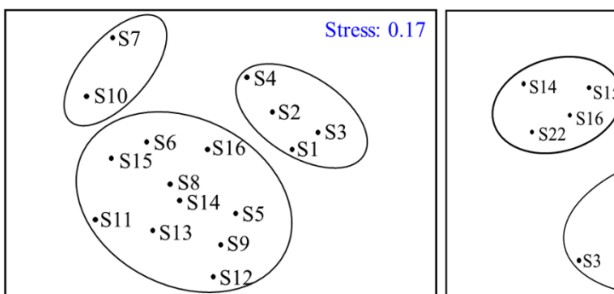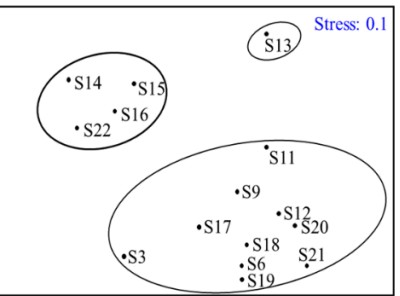

**Figure 3.** Results of clustering analysis of fish community sites from 2017 to 2018 (**A**) and from 2021 to 2022 (**B**) in Gehu Lake.

CLUSTER and NMDS were implemented to explore the fish populations of capture from 16 sites sampled in Gehu Lake from 2017 to 2018 (Figure 3A). Similarity, cluster analysis divided the sixteen sampling sites into three groups at a similarity level of 62.55%. The first group included S7 and S10, the second group included S1, S2, S3, and S4, and the third group included the rest of the sites. ANOSIM analysis further revealed that the composition of community structure was significantly different among the clustering groups (R = 0.781, $p < 0.05$). Meanwhile, the stress level was 0.17, suggesting a very good level of representation. The clustering result indicated that S14, S15, and S16, which were distributed in the northern part of Gehu Lake, were more dispersed. When compared, the fish population was more aggregated after removal. At the same time, the sampling sites in the southern seine area (S1, S2, S3, and S4) are better focused together.

### 3.4. Fish Composition in the Northern Conservation Area

The survey results in 2017–2018 and 2021–2022 show that in the northern part of the Gehu Lake reserve since the removal of the seine, *C. monggolicus* and *C. dabryi*, as well as *E. bambusa,* rose the IRI. The *C. alburnus* and *C. erythropterus* IRI declined, and the IRI of *C. nasus*, *Hemiculter leucisculus*, *H. bleekeri*, *Toxabramis swinhonis*, *Pseudorasbora parva*, *Rhinogobius giurinus*, and other small fish declined (Figure 4). The number percentages of these fish show the same result (Table 3).

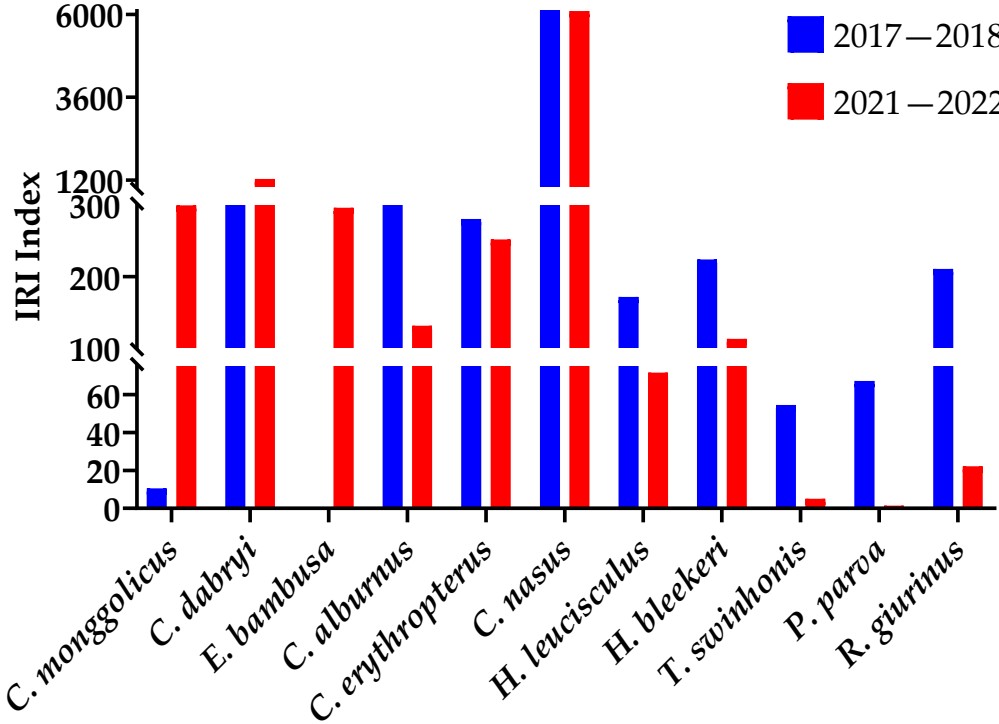

**Figure 4.** The IRI for carnivorous and small fish at the northern reserve in Gehu Lake in 2017–2018 and 2021–2022.

**Table 3.** The IRI and $N$% of fish in the northern conservation area of Gehu Lake in 2017–2018 and 2021–2022.

| | IRI | | $N$% | |
|---|---|---|---|---|
| Fish (Species) | 2017–2018 | 2021–2022 | 2017–2018 | 2021–2022 |
| *Coilia nasus* | 7048.78 | 6090.03 | 63.66 | 55.12 |
| *Hypophthalmichthys molitrix* | 6312.64 | 2588.34 | 7.71 | 2.10 |
| *Aristichthys nobilis* | 1602.66 | 3454.88 | 1.22 | 2.97 |
| *Carassius auratus* | 1250.46 | 1443.32 | 6.72 | 5.85 |
| *Culter dabryi* | 893.77 | 1226.79 | 2.82 | 7.60 |
| *Culter alburnus* | 573.32 | 130.55 | 1.91 | 0.97 |
| *Pseudobrama simoni* | 387.80 | 118.06 | 3.44 | 1.02 |
| *Pseudobagrus fulvidraco* | 282.55 | 194.24 | 2.21 | 1.17 |
| *Culterichthys erythropterus* | 280.84 | 251.88 | 1.30 | 1.27 |
| *Pelteobaggrus nitidus* | 240.74 | 371.31 | 2.06 | 3.22 |
| *Hemiculter bleekeri* | 223.53 | 112.09 | 2.06 | 1.46 |
| *Rhinogobius giurinus* | 210.27 | 22.04 | 0.46 | 0.29 |
| *Hemiculter leucisculus* | 171.22 | 71.49 | 1.45 | 0.93 |
| *Cyprinus carpio* | 93.80 | 744.07 | 0.69 | 1.32 |
| *Pseudorasbora parva* | 67.04 | 1.23 | 0.38 | 0.05 |
| *Toxabramis swinhonis* | 54.41 | 4.93 | 0.69 | 0.10 |
| *Misgurnus anguillcaudatus* | 53.69 | 1.36 | 0.53 | 0.05 |
| *Culter mongolocus* | 10.43 | 299.77 | 0.08 | 2.44 |
| *Ctenopharyngodon idellus* | 7.19 | 2.69 | 0.08 | 0.10 |
| *Acheilognathus macropterus* | 5.61 | 177.22 | 0.15 | 1.71 |
| *Pseudolaubuca engraulis* | 4.83 | 1.44 | 0.08 | 0.05 |
| *Pelteobagrus uachelli* | 3.14 | 2.35 | 0.08 | 0.05 |
| *Acheilognathus chankaensis* | 2.80 | 3.84 | 0.08 | 0.15 |
| *Sarcocheilichthys sinensis* | 2.72 | 0.00 | 0.08 | 0.00 |
| *Saurogobio dumerili* | 2.63 | 18.41 | 0.08 | 0.19 |

**Table 3.** *Cont.*

| Fish (Species) | IRI | | N% | |
| --- | --- | --- | --- | --- |
| | 2017–2018 | 2021–2022 | 2017–2018 | 2021–2022 |
| *Parabramis pekinensis* | | 674.88 | | 1.61 |
| *Paracanthobrama guichenoti* | | 336.13 | | 2.10 |
| *Elopichthys bambusa* | | 296.26 | | 0.63 |
| *Saurogobio dabryi* | | 250.86 | | 2.14 |
| *Xenocypris davidi* | | 190.78 | | 0.78 |
| *Hemibarbus maculatus* | | 182.26 | | 0.78 |
| *Plagiognathops microlepis* | | 146.51 | | 0.40 |
| *Megalobrama amblycephala* | | 123.97 | | 0.58 |
| *Megalobrama skolkovii* | | 44.41 | | 0.19 |
| *Cyprinus carpio × Carassius auratus* | | 19.77 | | 0.15 |
| *Sarcocheilichthys nigripinnis* | | 15.14 | | 0.19 |
| *Channa argus* | | 6.84 | | 0.05 |
| *Anguilla japonica* | | 3.32 | | 0.05 |
| *Siniperca chuatsi* | | 3.27 | | 0.05 |
| *Xenocyprisargentea* | | 2.20 | | 0.05 |
| *Paramisgurnus dabryanus* | | 1.82 | | 0.05 |
| *Abbottina rivularis* | | 1.23 | | 0.05 |

Notes: IRI. the Index of Relative Importance. *N%*. Number Percentage.

*3.5. Fish Composition in the Southern Seine Area*

The dominant species composition in the southern seine area of Gehu Lake in 2017–2018 and 2021–2022 is the same (Table 4). The main species are *C. nasus*, *H. molitrix*, *C. auratus*, *A. nobilis*, and *T. swinhonis*. After the removal of the seine, the IRI and the percentage of carnivorous fish increased, such as *C. dabryi*, *C. mongolocus*, and *C. erythropterus*, the number of *C. alburnus* fell, and the number of the newfound *E. bambusa* was also close to the common species level. In contrast, the IRI and number percentage of small omnivorous fish declined, such as *T. swinhonis*, *P. simoni*, *H. bleekeri,* and *H. leucisculus* (Figure 5).

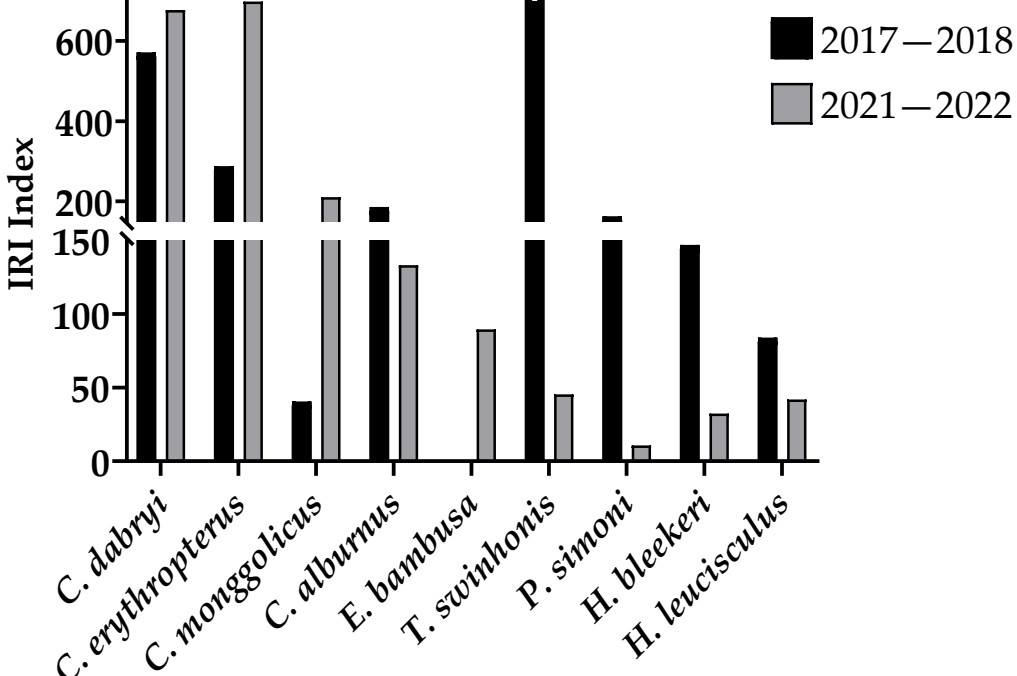

**Figure 5.** The IRI of carnivorous fish and small omnivorous fish in the seine area of the southern part of Gehu Lake in 2017–2018 and 2021–2022.

**Table 4.** Comparison of IRI and *N*% of fish in the seine area of the southern part of Gehu Lake in 2017–2018 and 2021–2022.

| | IRI | | *N*% | |
|---|---|---|---|---|
| **Fish (Species)** | **2017–2018** | **2021–2022** | **2017–2018** | **2021–2022** |
| *Coilia nasus* | 7336.27 | 8401.58 | 62.93 | 73.89 |
| *Hypophthalmichthys molitrix* | 4944.75 | 4619.04 | 7.51 | 5.22 |
| *Carassius auratus* | 1859.89 | 1011.72 | 4.32 | 3.75 |
| *Aristichthys nobilis* | 1181.84 | 2398.36 | 0.66 | 2.07 |
| *Toxabramis swinhonis* | 1011.33 | 45.39 | 11.37 | 0.40 |
| *Culter dabryi* | 570.46 | 677.08 | 1.31 | 2.59 |
| *Pseudobagrus fulvidraco* | 352.63 | 75.05 | 1.77 | 0.43 |
| *Cyprinus carpio* | 291.22 | 567.67 | 0.35 | 0.82 |
| *Parabramis pekinensis* | 291.07 | 246.34 | 0.58 | 0.30 |
| *Culterichthys erythropterus* | 287.21 | 698.26 | 1.31 | 3.32 |
| *Megalobrama amblycephala* | 195.64 | 286.63 | 0.27 | 0.43 |
| *Culter alburnus* | 185.25 | 133.38 | 0.77 | 0.46 |
| *Pseudobrama simoni* | 162.10 | 10.76 | 1.70 | 0.12 |
| *Hemiculter bleekeri* | 146.81 | 32.50 | 0.35 | 0.30 |
| *Hemiculter leucisculus* | 83.75 | 41.93 | 0.73 | 0.52 |
| *Culter mongolocus* | 40.42 | 210.54 | 0.15 | 1.86 |
| *Acheilognathus macropterus* | 37.05 | 53.67 | 0.39 | 0.49 |
| *Pseudorasbora parva* | 35.28 | 71.09 | 0.35 | 1.04 |
| *Pelteobagrus vachelli* | 32.75 | | 0.39 | |
| *Rhinogobius giurinus* | 31.36 | | 0.50 | |
| *Pelteobaggrus nitidus* | 28.73 | 93.45 | 0.39 | 0.73 |
| *Xenocypris davidi* | 17.40 | | 0.12 | |
| *Taenioides cirratus* | 7.67 | 6.94 | 0.12 | 0.09 |
| *Hyporhamphus intermedius* | 7.42 | 1.05 | 0.12 | 0.03 |
| *Misgurnus anguillicaudatus* | 6.75 | | 0.12 | |
| *Channa argus* | 6.21 | | 0.04 | |
| *Sarcocheilichthys* nigripinnis | 4.68 | 1.17 | 0.19 | 0.03 |
| *Siniperca chuatsi* | 4.40 | | 0.04 | |
| *Acheilognathus chankaensis* | 3.61 | | 0.08 | |
| *Pelteobagrus eupogon* | 2.17 | | 0.04 | |
| *Cobitis sinensis* | 0.86 | | 0.04 | |
| *Protosalanx hyalocranius* | 0.79 | 24.01 | 0004 | 0.21 |
| *Elopichthys bambusa* | | 89.69 | | 0.34 |
| *Anguilla japonica* | | 27.28 | | 0.12 |
| *Hemibarbus maculatus* | | 21.59 | | 0.06 |
| *Saurogobio dumerili* | | 18.72 | | 0.18 |
| *Paracanthobrama guichenoti* | | 5.61 | | 0.09 |
| *Megalobrama skolkovii* | | 2.03 | | 0.03 |
| *Saurogobio dabryi* | | 1.79 | | 0.03 |
| *Pseudolaubuca engraulis* | | 1.08 | | 0.03 |

Notes: IRI. the Index of Relative Importance. *N*%. Number Percentage.

Most bottom fish have declined, such as *Parabramis pekinensis*, *Misgurnus anguillcaudatus*, *Siniperca chuatsi*, *Sarcocheilichthys nigripinnis*, *Cobitis sinensis*, *P. eupogon*, *Pelteobagrus vachelli*, *Pseudobagrus fulvidraco*, *C. auratus*, and *P. simoni* (Figure 6).

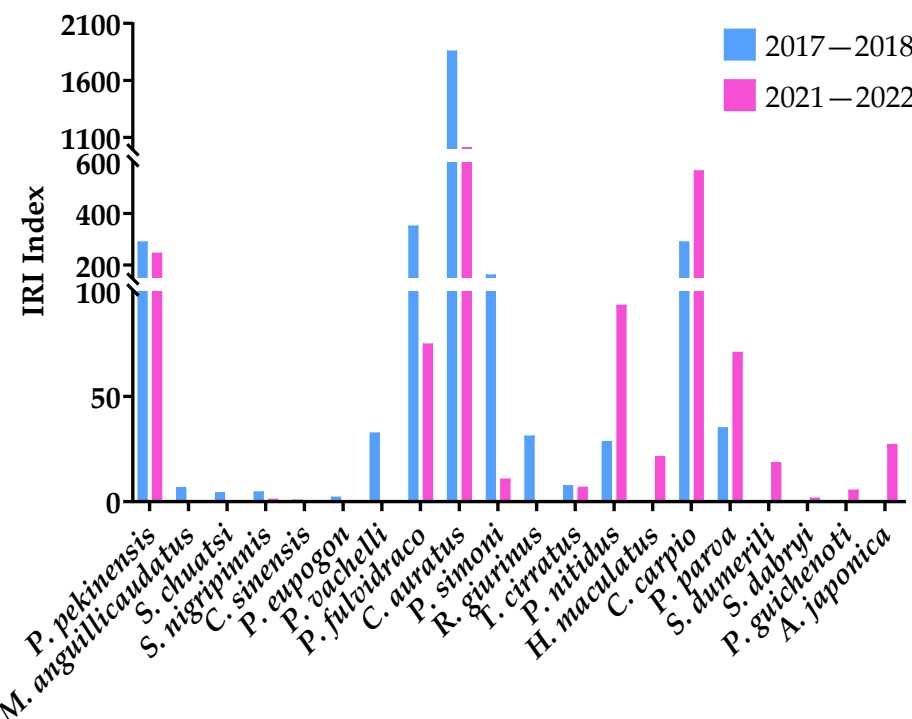

**Figure 6.** The IRI of bottom fish in the southern seine area of Gehu Lake in 2017–2018 and 2021–2022.

*3.6. Fish Composition in the Ecological Restoration Area*

The survey in 2021–2022 shows that the dominant species in the ecological restoration area of Gehu Lake (S11, S12, S13, and S21) are *C. nasus, H. molitrix,* and *A. nobilis*. Important species are *C. auratus, C. carpio, C. dabryi, C. erythropterus, P. pekinensis, T. swinhonis, H. leucisculus, P. fulvidraco, M. amblycephala, P. microlepis, E. bambusa,* and *Pelteobaggrus nitidus* (Table 5).

**Table 5.** The composition of dominant species and important species of fish in the ecological restoration area of Gehu Lake in 2021–2022.

| Fish (Species) | IRI | N% | W% |
|---|---|---|---|
| *Coilia nasus* | 9899.20 | 80.11 | 18.88 |
| *Aristichthys nobilis* | 3035.60 | 1.97 | 28.39 |
| *Hypophthalmichthys molitrix* | 2171.77 | 1.68 | 20.03 |
| *Carassius auratus* | 668.49 | 2.10 | 6.81 |
| *Cyprinus carpio* | 530.74 | 0.48 | 4.83 |
| *Culter dabryi* | 440.69 | 0.96 | 3.44 |
| *Culterichthys erythropterus* | 377.63 | 1.31 | 2.46 |
| *Parabramis pekinensis* | 360.05 | 0.37 | 3.23 |
| *Toxabramis swinhonis Günther* | 351.81 | 2.84 | 0.67 |
| *Hemiculter leucisculus* | 242.55 | 2.30 | 0.13 |
| *Pseudobagrus fulvidraco* | 231.08 | 0.77 | 1.54 |
| *Megalobrama amblycephala* | 212.72 | 0.28 | 1.84 |
| *Plagiognathops microlepis* | 203.97 | 0.26 | 1.78 |
| *Elopichthys bambusa* | 165.08 | 0.22 | 1.98 |
| *Pelteobaggrus nitidus* | 152.51 | 1.18 | 0.34 |

Notes: IRI. the Index of Relative Importance. *N%*. Number Percentage. *W%*. Weight percentage.

*3.7. Community Stability*

An analysis of the ABC curve results showed that the curves for the whole lake in 2021–2022 showed a large crossover with a W value of −0.044 (Figure 7B), indicating that the fish community is in a moderate disturbance and the degree of disturbance increased compared

to 2017–2018 (Figure 7A). The ABC curves of the two survey results in the northern region showed little change and were lightly disrupted (Figure 7C,D). A comparison of the ABC curves for the remaining lakes reveals that the lakes are slightly disturbed in 2017–2018 (Figure 7E) and moderately disturbed in 2021–2022 (Figure 7F).

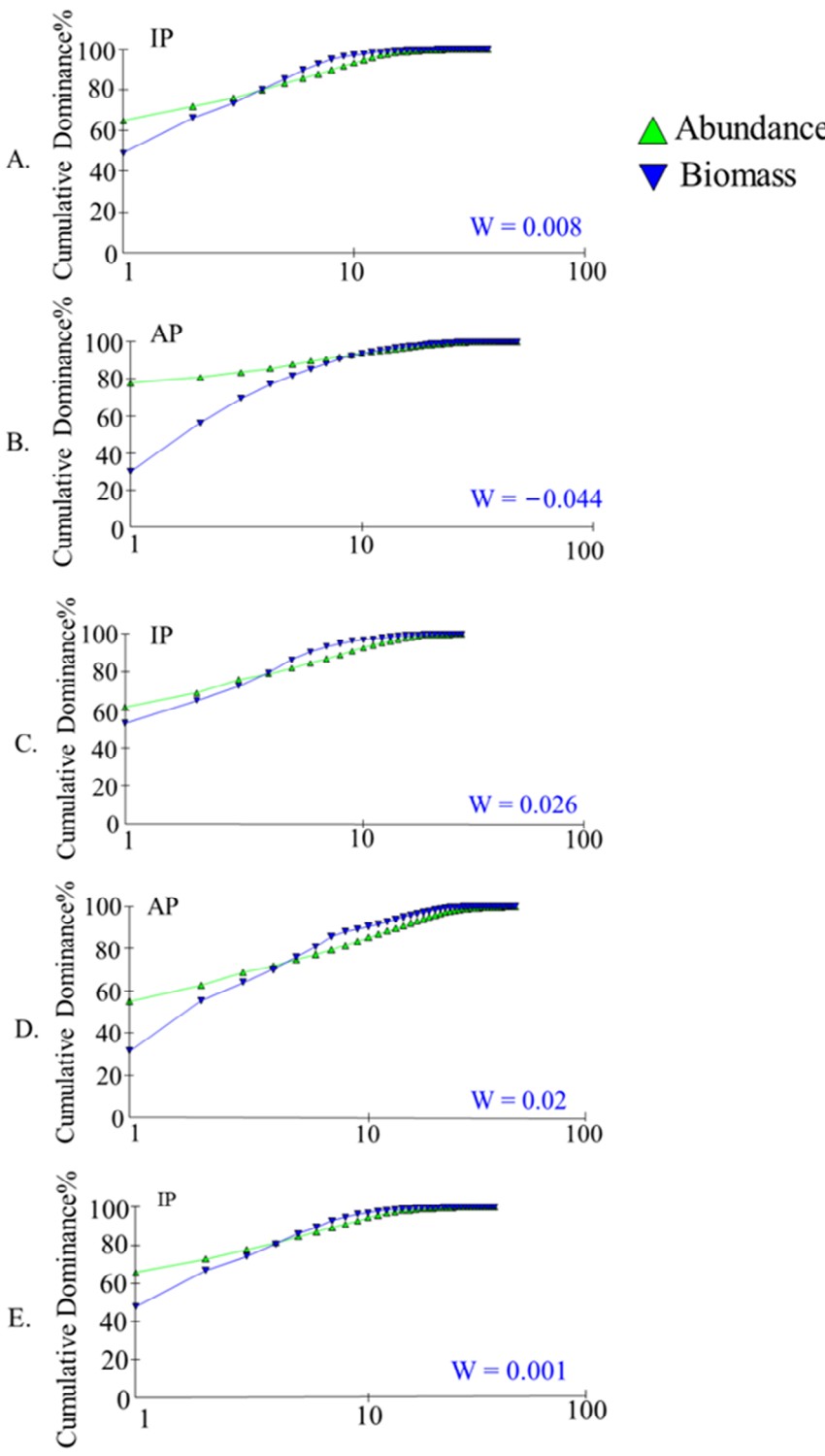

**Figure 7.** *Cont.*

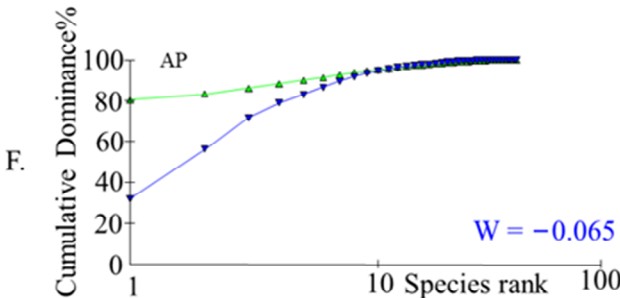

**Figure 7.** Comparison of ABC curve results in different lake areas of Gehu Lake. (**A,B**) represent the results for the entire lake area, (**C,D**) represent the results for the northern protected area, and (**E,F**) represent the results for the lake area outside the northern protected area. IP represents the sampling results from 2017 to 2018 and AP represents the sampling results from 2021 to 2022.

*3.8. Community Diversity*

As can be seen in Table 6, after the removal of net enclosures, the $H'$-index of the northern protected region rose, while that of the rest part and the seine area in the southern part of Gehu Lake declined and, for the whole lake, the $H'$-index declined. $J'$-indexes decreased comprehensively in this study compared to that in 2017–2018. The $R'$-index of the northern protected region rose after removal, while that of the rest of the parts and the seine area in the southern part of Gehu Lake declined and, for the whole lake, the $R'$-index rose.

**Table 6.** Comparison of fish diversity indices in different areas of Gehu Lake.

| | $H'$ | | $J'$ | | $R'$ | |
|---|---|---|---|---|---|---|
| Years | 2017–2018 | 2021–2022 | 2017–2018 | 2021–2022 | 2017–2018 | 2021–2022 |
| TL | 1.576 | 1.194 | 0.127 | 0.072 | 4.144 | 4.615 |
| NL | 1.656 | 2.020 | 0.202 | 0.184 | 3.401 | 5.245 |
| RL | 1.528 | 1.039 | 0.132 | 0.079 | 3.889 | 3.637 |
| SL | 1.537 | 1.258 | 0.145 | 0.113 | 3.943 | 3.706 |

Notes: $H'$. Shannon–Weiner diversity index. $J'$. Pielou evenness index. $R'$. Margalef's index. TL. the total lake. NL. the northern lake. RL. the remaining lake except for the northern protected area. SL: the southern lake in the seine area.

## 4. Discussion

*4.1. Species Composition*

Gehu Lake is a shallow lake, in which the fish community composition is dominated by sedentary fish and influenced by artificial fishing and pen culture to a great extent. According to the survey in the 1950s, 12 orders, 21 families, and more than 60 fish species were living in Gehu Lake [28]. Nevertheless, only thirty species of fish in seven orders and nine families were reported in 2008 [7] and, in 2017–2018, a total of thirty-six species of fish in four orders and eight families were collected in Gehu Lake [29]. This survey was conducted for the first time after all net enclosures were removed in 2019, and a total of fourty-four species of fish in seven orders and ten families were collected. In addition, an alien species (*Cirrhinus mrigala*) and a hybrid species (*Cyprinus carpio × Carassius auratus*) was detected. The additions consist mainly of migratory fish and include *E. bambusa*, which has not been surveyed previously but has become common-specific at this time. The number of fish species increased, as that of carp increased from 26 to 34, and accounted for a greater share of the total. Fish species composition and quantity could directly reflect the change in characteristics of the fish community. After the removal of the net enclosure, a significant change in fish species number and composition occurred in Gehu Lake.

The introduction and use of non-native species play one of the most critical roles in the rapid development of the aquatic industry [30]. The majority of non-native fish are introduced voluntarily for the aim of fisheries development, and they play an important role

in guaranteeing food security [31]. However, non-native fish have become a widespread hazard to native fish across the world, and measures are needed to manage invasive fish while safeguarding native species.

Gehu Lake is located north of the Yangtze River, east of Taihu Lake, and west of Changdang Lake. The well-developed network of water systems and intensive human activities, coupled with the high dispersal ability of alien aquatic animals, result in aquatic ecosystems that are vulnerable to disturbance by invasive alien species. *C. mrigala* was discovered as an alien species in Gehu Lake for the first time in this survey, which is native to India, Bangladesh, and other locations, and its introduction into our nation was first used as a food fish [32]. When the consumer market shrank, *C. mrigala* was employed as bait for carnivorous species in the southern area. In recent years, due to abandonment and escape during the breeding process, a large number of *C. mrigala* have entered natural waters and flourished in the lake, competing for food and space with indigenous fish and causing long-term harm to the lake ecosystem [33]. Because Gehu Lake lies closer to a residential area and the northern part is a parkland scenic region, *C. mrigala* is more likely to be the product of random abandonment by the nearby population. Indeed, there are numerous cases of alien fish invasion. For example, the number of alien fish species in the Dongguan section of the Dongjiang River, one of the main streams of the Pearl River system, has been increasing year by year and, in 2015, Tilapia and *C. mrigala* became the dominant species of fish resources, with entry routes including blind release, farming escape, and random abandonment [34]. The survey also included the tetraploid hybrid fish species *C. carpio* × *C. auratus*, which was created by the Institute of Aquatic Biology of the Chinese Academy of Sciences and has some commercial significance. The investigation discovered that *C. carpio* × *C. auratus*s was from the northern part of Gehu Lake, the majority of which are park scenic areas that are specially set up to release aquatic animals in the water in the scenic area. Therefore, there is a possibility of blind release by local residents in the case of not understanding the type of fish, the removal of the net enclosure to further expand the living space and, to some extent, to increase the survival rate of released fish. While the introduced alien fish meet the protein demand and boost the market economy, the invasion of a small portion of alien fish can cause some damage to the global aquatic ecosystem, especially after the removal of net enclosures, which on the one hand promotes the enhancement of habitat connectivity of local species but, at the same time, increases the potential for the movement and spread of non-native species [35].

The removal of the enclosure nets has the potential to increase the risk of invasion of exotic species (farm escapes and anthropogenic releases) into Gehu Lake. China is the largest freshwater aquaculture country in the world, and many of the areas are set near natural waters. The risk of invasive alien species in many lakes in China has increased significantly since 2020 [36], probably due to the failure to regulate farming activities and the removal of seine nets in the corresponding natural lakes, which has implications for other lakes in terms of the management of pen culture. So, it is necessary to strengthen the scientific management of exotic fish in fisheries management measures to promote the development of aquaculture and the ecological safety of lakes, as well as to do a good job of popularizing alien species to the surrounding residents and farmers.

### 4.2. Dominant Species

Whether it is cage culture, net culture, or pen culture, there is an aggregation effect of fish due to the accumulation of residual bait and feces during the culture process, which enriches the nutrients and thus attracts detritivorous and zoobenthivorous fish to feed [37]. As pen culture encroaches on the habitat and food resources of other fish species, when the net enclosure is removed, this part of the encroached spatial and nutritional niche is released, which inevitably triggers competition from other fish species. From the results, the IRI of carnivorous fish such as *C. nasus*, *C. dabryi,* and *C. mongolicus*, and herbivorous fish such as *M. amblycephala* and *P. pekinensis* increased, indicating that they won the competition for spatial and trophic niches. This indicates, to some extent, that the removal of enclosure

nets improved the survival environment of these fish and expanded their feeding range, and also improved the protection of spawning grounds in the natural environment which was beneficial to the recovery of their populations.The IRI of baitfish such as *T. swinhonis*, *H. bleekeri*, *P. fulvidraco,* and *P. nitidus* decreased, indicating that they were at a disadvantage in the competition.

According to R/K selection theory [38], after the net enclosures were removed, the short-cycle growing fish, such as *C. nasus*, had quicker growth and greater reproductive capacity in the early stage and faster resource recovery in the later stage, whereas the long-cycle growing fish had delayed recovery and faster recovery in the latter stage due to a shortage of food in the first stage. With the change in diet from zoophytoplanktivores to carnivorous, the width of the food niche has become wider. At the same time, the fast growth and high fertility of the *C. nasus*, coupled with the implementation of the retreat policy, has led to an explosive increase in the number and size of *C. nasus* in the last two years, which has reduced the survival space of other small fish. *C. dabryi* and *C. mongolicus* are top predators in aquatic ecosystems and, in the absence of fishing, populations expand and feed heavily on smaller fish, leading to declines in the numbers of *T. swinhonis*, and *H. bleekeri*. After the removal of the net enclosure in Gehu Lake, the situation of the quickly expanding small fish, such as *C. nasus*, will encroach on the living area of other fish. This is not conducive to the stability of the lake ecosystem and should be governed by reasonable management and control.

O. rubicundus and *P. eupogon* were not discovered in this survey when the species of the captures were compared to the survey data from 2017 to 2018. O. rubicundus is an estuary fish that spends the majority of its life cycle in environments that are semi-saline, have sand and gravelly bottoms, and have clear water [39]. The removal of the net enclosure temporarily disrupted the substrate, which had an impact on the water quality and reduced their populations [40,41]. The Pelteobagrus genus is a native fish of Gehu Lake, its natural production was initially low, and the IRI of both *P. fulvidraco* and *P. nitidus* declined when the net enclosures were removed because of the disappearance of the cluster effect. *P. eupogon*, which is categorized as vulnerable on the Red List of Chinese species [42], originally had a tiny population in Gehu Lake, and the population fell when the net enclosure was removed, most likely owing to increased competitive pressure for food and greater vulnerability to predation by natural predators [43], which made it difficult to catch.

It is worth noting that 37 *E. bambusa* were captured in this survey, yet none of which were collected before removal in 2017–2018. Compared with other fish species, the average weight of *E. bambusa* was higher, resulting in an IRI greater than 100 (IRI = 109.35). Additionally, this species was captured in 11 of the 15 sample sites, indicating the wide distribution in Gehu Lake. Hence, one can see that, to some extent, the removal of the net enclosure contributes to the existence and growth of the *E. bambusa* population. *E. bambusa*, also known as "water tiger", is a big fierce fish species that feed on other fish. In the past, it was referred to as a pest in aquaculture and needed to mainly eliminate in natural fish fry production [44]. However, as a fierce carnivorous fish, *E. bambusa* occupies a high trophic ecological level in the lake, and benefits in eliminating the sick and weak individuals and inhibiting the growth of small fish species within the range of moderate quantity, which is instrumental in maintaining the stability of the water ecosystem [45]. Nevertheless, when exceeding a certain amount, it may do harm to the fish community structure and fishery resources in the lake, which necessitates continuous investigation and reasonable control.

### 4.3. Community Structure and Stability

The cluster analysis from 2021 to 2022, which revealed that sites S14, S15, S16, and S22 formed a more aggregated area with higher similarity in fish community composition while the other areas had a lower similarity, explained that the structure of the fish community in this aggregated area differed from other lake areas. At the same time, the ABC curve data suggest that the region was little disturbed in both survey periods, indicating that this

area was less disturbed by humans and the overall condition is more stable. According to the relevant information, the National Ministry of Agriculture designated this region as the Culter national aquatic germplasm resources protection zone of Guhu Lake in order to accomplish the conservation and wise use of Gehu Lake fish genetic resources and their survival habitat. The main species to be protected are *C. alburnus*, *C. dabryi*, and *C. mongolicus*. Because of their early establishment in 2009 and the commencement of the retreat policy in 2018, they has been allowed a longer natural recovery period. The three species of Culter are all carnivorous fish. *C. dabryi* and *C. alburnus* mainly feed on *C. nasus*, while *C. mongolocus* feeds on *P. parva*, *H. leucisculus*, and *T. swinhonis*. The removal of seine nets has improved the living environment of Culter populations to a certain extent and expanded their feeding range; the growth of "predator" populations has led to a reduction in "prey" fish populations (Figure 4). However, the decline in the number of *C. alburnus* and *C. erythropterus* may be related to the large increase in the number of *E. bambusa*, which also prey on other small fish [20]. Additionally, compared to other parts of Gehu Lake, the northern area of Gehu Lake's shoreline is mostly made up of urban and natural parks, which not only have the capability to enhance water quality and ecological control, but also provide a lesser risk of external pollutant intrusion [46]. Compared to the study in 2017–2018, H′-index grew in the northern reserve, indicating that the structure of the fish community has become more complicated [23]. The fish community structure of the northern section of the Gehu Lake reserve exhibits a better improvement condition and its structure is more stable through a series of fisheries management actions, such as the removal of net enclosure and retreat.

The number of small fish in the pen culture area in the southern region of Gehu Lake decreased significantly after the removal of the seine, which is inconsistent with the results of Gu et al. [13] in East Taihu Lake, which may be related to the number of carnivorous fish. The southern seine area of Gehu Lake in 2017–2018 (S1, S2, S3, S4, and S5) focused on the culture of *H. molitrix* and *A. nobilis*, so they occupy a larger spatial niche. Carnivorous fish such as *C. dabryi*, *C. alburnus*, and *C. erythropterus* have reached the level of important species. After the removal of the seine and before *H. molitrix* and *A. nobilis* that occupied the spatial niche were released [47], the predation range of carnivorous fish expanded and the number of fish species that can be predated increased, In addition, the original seine area is rich in bait residue, which attracts omnivorous and carnivorous fish to congregate here. Thus, the number and IRI of fish such as *C. erythropterus*, *C. dabryi*, *C. mongolocus*, *C. carpio*, and *E. bambusa* increased. With the growth of the carnivorous fish population such as *T. swinhonis*, *P. simoni*, *H. bleekeri*, and *H. leucisculus*, which should have increased in number, have declined because they were predated. Gu et al. mentioned that the bait residues in the original seine culture area attracted omnivorous fish to gather, and it so happened that the dominant species composition of fish in the area was *C. auratus*, *H. leucisculus*, *Acheilognathus macropterus*, *P. fulvidraco*, *Pseudorasbora parva*, *A. nobilis*, and *C. nasus*, while the number of carnivorous fish was poor. Table 6 shows that the biodiversity, in general, decreased after the removal of the seine, which may be related to the fluctuation of water quality due to the removal of the seine [48] and also to the competition among fish populations for new space [49,50].

After the removal of the seine in 2019, the local government in the central area of Gehu Lake (S11, S12, S13, and S21) established "the fish-controlled algae" ecological restoration area with floating net interception facilities. The southeast wind in the summer will make cyanobacteria blow to the northern part of Gehu Lake. Thus, the central lake area will establish an ecological restoration area, mainly by non-baiting farming of *H. molitrix* and *A. nobilis*, to control the number of cyanobacteria, reduce the degree of eutrophication of the water body, and achieve the purpose of repairing and improving water quality. *H. molitrix* and *A. nobilis* from the southern seine culture area are now transferred to the central ecological restoration area of the lake, and the spatial niche of the former seine culture area ecosystem in the south is released, while the new *H. molitrix* and *A. nobilis* populations seize the spatial niche of the central ecological restoration area, making the spatial niche

of other fish populations in the area shrink. The proliferation of *H. molitrix* and *A. nobilis* in the seine may have an impact on the overall composition of the fish community. Zhao et al. used stable isotope techniques to conclude that the excretion of *H. molitrix* and *A. nobilis* in the Meiliang Bay seine in Taihu Lake caused significant changes in sediment composition and that the fish community inside the net may have higher stability than outside the net [36]. The stocking of *H. molitrix* and *A. nobilis* will certainly have some impact on the ecosystem of the Gehu Lake ecological restoration area, so more in-depth research is subsequently needed.

According to the ABC curve for the entire lake, the fish community structure is in a moderate anthropogenic disturbance condition in 2021–2022, and is highly disturbed and less stable compared to 2017–2018. When the fish community structure is disrupted at a larger level, the composition of fish species will eventually be dominated by small, fast-growing species [38]. As shown in Table 3, the increase in IRI of *C. nasus* and other small fish indicates that the small fish have exhibited a growing trend in the lake in recent years, while their H′ lowers, implicitly indicating that the lake is now in a weak stable condition. The decrease in both the W value of the ABC curve and the value of the H′-index for the remaining lakes revealed that the lakes were not as stable in 2021–2022 as they were prior to the entire removal of the net enclosures in 2017–2018. The lakes went from a more stable to a weakly stable condition after the net enclosures were completely removed. Lakes can achieve a certain degree of stability owing to human management involvement during pen culture, but the removal of net enclosure facilities influences the living environment and spatial distribution of fish, eliminating human management to enable natural recovery, and thus a weak stable state recovery phase is normal.

After removal, changes in fish habitat led to changes in biodiversity, while related studies show that changes in habitat will have an impact on biodiversity. According to Edge et al., changes in habitat were to blame for the decline in the α-diversity of fish populations in rivers [51]. Dam building, similar to the net enclosure, can obstruct the passage of fish disrupt the environment of fish populations, resulting in a decline in biodiversity [52]. Subsequent ecological monitoring of Gehu Lake will continue, as well as strengthening fish habitat protection, thus maintaining the balance of the ecological system of the lake.

## 5. Conclusions

This study, for the first, time reveals the alteration of the fish community in Gehu Lake after the removal of the net enclosure. The results revealed an increase in the number of fish species compared to before the removal of the net enclosures, with five new migratory fish species discovered: *A. japonica*, *M. skolkovii*, *X. microlepis*, *C. mrigala*, and *E. bambusa*. In this survey, *P. eupogon*, an animal identified as vulnerable on the Red List of Chinese Species, was not discovered. The common fish species have changed significantly, with a general tendency toward more small-sized fish. Overall, the stability of the fish community in Gehu Lake decreased after removal, while that in the northern protection zone increased. The fish community in the southern seine area has undergone great changes and the restoration effect of the ecological restoration area is worth continuing to monitor. The removal of the net enclosure is a significant anthropogenic disturbance reduction in lake management endeavors that urges continued monitoring to determine its long-term ecological consequences.

**Author Contributions:** D.X., S.J. and X.R. conceived and designed the experiments, X.R. and S.J. were responsible for data scoring, analyses, and writing the manuscript. Y.L. and D.F. helped select the samples, L.R. and D.X. helped with data analysis during manuscript preparation. All authors have read and agreed to the published version of the manuscript.

**Funding:** This research was funded by the National Key Research & Development Program of China (2020YFD0900500), the Central Public-interest Scientific Institution Basal Research Fund, CAFS (No. 2020TD61 and No. 2020XT13), and the Central Public-interest Scientific Institution Basal Research Fund, Freshwater Fisheries Research Center, CAFS (No. 2021JBFM17).

**Institutional Review Board Statement:** This study was approved by the Care and Use Committee of the Ministry of Freshwater Fisheries Research Center of the Chinese Academy of Fishery Sciences. All animal experimental procedures followed the Guideline for the Care and Use of Laboratory Animals in China.

**Data Availability Statement:** The data sets generated during and/or analyzed during the current study are available from the corresponding author upon reasonable request.

**Conflicts of Interest:** The funders had no role in the design of the study; in the collection, analysis, or interpretation of data; in the writing of the manuscript, or in the decision to publish the results.

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
