# Peer review of "Changes in Fish Assemblage Structure after Pen Culture Removal in Gehu Lake, China"

_fishes, doi:10.3390/fishes7060382_

Round 1
Reviewer 1 Report
1. The article aims at comparing fish diversity data from Gehu Lake before and after pen culture removal. However, no information is provided on the methodologies used in 2017-2018. Are the differences observed between 2017-18 and 2021-22 due to methodological differences? Without this information, it is impossible to assess the validity of the conclusions that were made.
2. Despite the answer provided to the previous question, I urge the authors to provide details of the sampling methodologies performed in 2017-2018 since the article shows data for this period. Overall, the 2017-2018 sampling is part of this study and should be described in detail in the methods.
3. Although the article's message is clear, there are numerous opportunities to improve the grammar and style of the article. You can try using the free version of Grammarly.com that allows correcting some grammar and style in word files.
4. Pay close attention to minor errors, like species not written in italics, beginning a sentence without a capital letter, missing commas, etc.
5. The font size in figures 1 and 2 (the coordinates in both maps) is too small to read.
6. Figure 3- The nMDS plot for 2017-2018 should be on the left, and 2021-2022 should be shown on the right. Write down above each panel what it corresponds to - "before pen removal", "after pen removal". The font size is also too small.
7. Figure 7- images are blurry and the font size is too small. There is no need to have a legend in each panel since it is the same for all panels. Place the W data at the bottom right of the plot and not in the middle.
8. Line 283, when you mention foreign fish I suspect that you are referring to a non-native species. If so, please correct it.
9. Since you delve so much into non-native and invasive species in the discussion, I urge including a reference to the native-non-native-invasive status of each species in table 1. You can add a new column for this (Native- Nat., Non-native- Non-nat, Invasive- Inv.) (use the abbreviations in each line). To make space for this new column, you can abbreviate the functional guild of each species; for example, zoophytoplanktivores can be referred to planktivores and abbreviated to plankt., phytoplanktivores can be abbreviated to phytoplankt. Please mention in the table's legend the meaning of each abbreviation, including the meaning of FFG.
10. Do not separate the main conclusions with ; but rather with a . since you begin each sentence with a capital letter.
Reviewer 2 Report
First of all, papers with very similar subjects are already published, e.g. the paper cited in the manuscript Gu, X. k.; Shen, D. D.; Gui, Z. Y.; et al. „Study on Community Structure and Biodiversity of Fish in East Taihu Lake after Removal of Net Enclosure“. Journal of Aquaculture. 2022, 43 (2), 1–9. Thus, I do not see much sense to publish some similar results, especially those that are obtained from the same area, as then the question is raised: what is the benefit of such a paper for general knowledge?
However, even if the approach would be beneficial in the sense that this manuscript is having some more additional data that can be important for other regions, thus worth publishing, unfortunately this manuscript suffers from some serious flaws in designing the study. As the main focus is to compare the data obtained before pen culture removal and after, for a proper comparison all the possible impacting factors need to be the same. However, by comparing sampling sites from this manuscript and sampling sites indicated in Li et al. 2022-„Spatial and temporal distribution of fish assemblages and its relationship with environmental factors in Lake Gehu“ for the period 2017-2018 it can be seen that number of sites is not the same in both studies and, what is even more important, the actual sampling sites are distributed unevenly, e.g. the southern side of the lake has at least twice as more sampling sites in 2017-2018 compared to new sites. Consequently, it is impossible to get a precise comparison between periods if the sampling method (including effort) and the area are different. Furthermore, many long term changes were already described in that paper (Li et al., 2022), thus, the change that occurred between a such short period of time, before and after the pen culture removal, does not seem so important as it can be presumed that the real change could be determined only after a longer period of time then this study encompass. Hence, probably because of different sampling factors the authors failed to convincingly explain the fluctuation in species number between the periods, e.g. 60 species in 1950s, only 30 species in 2008, 36 species in the period 2017-2018 compared to 44 species during the current study. Explanations used in a manuscript do not correspond to the short period of 2 years after the removal of net enclosures as many other questions are immediately raised that cast a doubt on meaningful explanations used by the authors. Thus, taking into account the aforementioned, the authors are advised to restructure the manuscript in a manner that will focus more on the current community structure and fish biodiversity and to submit it to the journal with a more regional focus.
Reviewer 3 Report
Ren et al. has analysed the community structure of the Gehu Lake after the removal of the net enclosure. This study, for the first time, reveals the alteration of fish community and an increase in the number of fish. Until now, only few papers have documented the modification of the community structure after removal of the net enclosure in lakes.The interest in patterns of lake diversity has been rising recently. The new information can be very valuable and increase the existing knowledge concerning the composition of dominanting species.
Material and methods
Some aspects of the data analysis: 3.5. Community clustering characteristics need to be specified and extended (softwares, packages, languages, etc, used).
Round 2
Reviewer 1 Report
Dear colleagues,
Thank you for going providing a revised version of your article. Now, I only have minor changes to suggest.
1. Fig. 2 - The locations of the sampling sites are not the same in the two sampling periods. So, keep the same labeling for those that were sampled in both periods: for example, S13, S14, and S15 (2017-2018) seem to be in the same location as S1, S2, and S3 (2021-2022), while S4 (2021-2022) is a new sampling station. Also, start the labeling of 2017-2018 from the bottom of the map, with S12 now being station S17 since S16 was the last station label for 2017-2018.
2. Fig. 4 - Keep "species rank" only in the bottom plot, all the others can be removed from the upper panels.
3. Conclusions - add space between the genus and species.
Reviewer 2 Report
I would like to thank the authors for their reply to my comments but I'm afraid that after reading it and checking only minor corrections that have been made in the manuscript, my primary concern stays the same: I do not see the benefit of publishing those results for a wider scientific community. Point is that certainly specific data related to Gehu Lake are unique as it represents the particular habitat with its own biological and ecological characteristics, thus results will describe these specific changes that occurred within the fish community in Gehu Lake after the removal of net enclosure. However, the main point is that nothing unusual happened in comparison to the other lakes/habitats after the removal of net enclosures as such a result has already been described in other areas/lakes and published papers. Thus, publishing the paper in Fishes needs more significant results and conclusions, than those presented in this paper, which would change or increase the current knowledge related to the subject in the case. However, the major paradigm is the same and these results confirmed something already known from other regions, even in the vicinity of the Gehu Lake. Publishing something that has the same hypothesis, methodology, general results and conclusions as previously published studies makes no sense (for a journal with such high IF as Fishes) as eventually every other study performed in any other lake could be published just because the fish species are bit different than any other.
Thus, to conclude, I do not recommend publishing such papers.
